# An Sfi1-like centrin-interacting centriolar plaque protein affects nuclear microtubule homeostasis

Christoph Wenz[1], Caroline Sophie Simon[1,¤a], Tatiany Patricia Romão[1,¤b], Vanessa Saskia Stürmer[1], Marta Machado[1,2], Natacha Klages[3], Anja Klemmer[1], Yannik Voß[1], Markus Ganter[1], Mathieu Brochet[3], Julien Guizetti[1] *

1 Center for Infectious Diseases, Heidelberg University Hospital, Heidelberg, Germany, 2 Graduate Program in Areas of Basic and Applied Biology, Instituto de Ciências Biomédicas Abel Salazar, Universidade do Porto, Porto, Portugal, 3 Department of Microbiology and Molecular Medicine, Faculty of Medicine, University of Geneva, Geneva, Switzerland

☯ These authors contributed equally to this work.
¤a Current address: European Molecular Biology Laboratory, Advanced Light Microscopy Facility, Heidelberg, Germany.
¤b Current address: Department of Entomology, Instituto Aggeu Magalhães-FIOCRUZ, Recife, Brazil.
* julien.guizetti@med.uni-heidelberg.de

**Data Availability Statement:** All relevant data are within the manuscript and its Supporting Information files.

## Abstract

Malaria-causing parasites achieve rapid proliferation in human blood through multiple rounds of asynchronous nuclear division followed by daughter cell formation. Nuclear divisions critically depend on the centriolar plaque, which organizes intranuclear spindle microtubules. The centriolar plaque consists of an extranuclear compartment, which is connected via a nuclear pore-like structure to a chromatin-free intranuclear compartment. Composition and function of this non-canonical centrosome remain largely elusive. Centrins, which reside in the extranuclear part, are among the very few centrosomal proteins conserved in *Plasmodium falciparum*. Here we identify a novel centrin-interacting centriolar plaque protein. Conditional knock down of this Sfi1-like protein (PfSlp) caused a growth delay in blood stages, which correlated with a reduced number of daughter cells. Surprisingly, intranuclear tubulin abundance was significantly increased, which raises the hypothesis that the centriolar plaque might be implicated in regulating tubulin levels. Disruption of tubulin homeostasis caused excess microtubules and aberrant mitotic spindles. Time-lapse microscopy revealed that this prevented or delayed mitotic spindle extension but did not significantly interfere with DNA replication. Our study thereby identifies a novel extranuclear centriolar plaque factor and establishes a functional link to the intranuclear compartment of this divergent eukaryotic centrosome.

## Author summary

Most fatal malaria cases are caused by the unicellular parasite *Plasmodium falciparum*. The parasite invades red blood cells where it grows and replicates to ultimately release up

**Funding:** The German Research Foundation (DFG) (349355339) funded J.G. and provided salary for C.S.S. The Human Frontiers Science Program (CDA00013/2018-C) funded J.G. and provided salary for A.K. and V.S.S. The Daimler and Benz Foundation funded J.G. The Chica and Heinz Schaller Foundation funded the salary of J.G. The "Studienstiftung des Deutschen Volkes" funded the salary of Y.V. The "Landesgraduiertenförderung Baden-Württemberg" funded the salary of C.S.S. The German Research Foundation (DFG) – Project number 240245660 - SFB 1129 and the Baden-Württemberg Foundation (1.16101.17) funded M.G. The Fundação para a Ciência e Tecnologia (FCT, Portugal) - PD/BD/128002/2016 funded the salary of M.M. Collaborative work between the laboratories of J.G. and M.B. was possible with the support of an EMBO short-term fellowship (8314) to C.S.S. Work in M.B. laboratory is supported by the Swiss National Science Foundation (31003A_179321 and 310030_208151). The funders had no role in study design, data collection and analysis, decision to publish, or preparation of the manuscript.

**Competing interests:** The authors declare that they have no conflict of interest.

to 30 new daughter cells from the bursting erythrocyte. Its proliferative cycle causes an exponential increase of parasite biomass in the patients and depends on an atypical cell division mode. Contrary to most model organisms the parasite undergoes several rounds of nuclear multiplication prior to formation of multiple daughter cells at once. The parasite centrosome, called centriolar plaque, thereby plays an essential role. It coordinates nuclear divisions by organizing an array of microtubules, called mitotic spindle, that segregates chromosomes. Yet, the centriolar plaque remains poorly characterized. Here we reveal the second known component of the extranuclear centriolar plaques compartment. Knock down of this Sfi1-like protein, PfSlp, causes aberrant mitotic spindles and results in a failure to properly divide the nuclei. Our findings suggest that the extranuclear and the intranuclear compartment of the centriolar plaque are functionally linked. We reveal more details about the divergent cell division mode of this deadly pathogen that might be exploited for future intervention strategies.

## Introduction

Cell division drives the expansion of life, which includes infectious pathogens, such as *Plasmodium falciparum* that proliferate in the blood of their human host. This parasite can thereby cause malaria which still results in more than 600,000 deaths per year [1]. Multiplication within the red blood cell that is invaded by the intracellular parasite is driven by an atypical cell division mode, called schizogony [2,3]. The beginning of the schizont stage is not precisely defined but occurs around 28 hours post invasion (hpi) once *P. falciparum* has gone through the preceding ring and trophozoite stages [4]. Schizogony consists of multiple rounds of nuclear division without cytokinesis yielding between 8 to 30 nuclei. At about 44 hpi those nuclei are packed into the daughter merozoites that egress from their host cell after about 48 hpi. How the parasite "decides" when the final number of nuclei is reached is unclear, but some studies suggest an implication of nutrients [5–7]. These cycles of nuclear division, merozoite formation, egress, and reinvasion can result in the high parasitemias that are associated with severe disease progression [8].

Our understanding of cell division in *Plasmodium* spp. is increasing but due to the high divergence from model organisms many of the underlying molecular mechanisms remain unclear [9,10]. Notable differences are the autonomous nuclear cycles of DNA replication, chromosome segregation, and division [11,12]. During this process chromosomes are not condensed, and the nuclear envelope remains intact. The lack of a genetically encoded nuclear envelope marker has limited a more detailed study of karyofission but recent adaptation of membrane staining will allow progress in this domain [13,14]. Classical components of cell cycle regulatory pathways, such as cyclins, kinases, and phosphatases, are significantly less conserved [9,10,15–21].

Whether cell cycle checkpoints similar to the ones described in model organisms exist in the malaria-causing parasite is still debated [22–25]. The observation that drug-mediated depolymerization of microtubules, which would trigger a spindle assembly checkpoint, does not cause a noticeable delay in DNA replication has been interpreted as the absence of this checkpoint [24,26]. Depletion of components of the mini-chromosome maintenance complex despite leading to aberrant spindles has also been speculated not to trigger a checkpoint [27]. Conclusive and time-resolved data showing an inducible and reversible delay in cell cycle progression, which is necessary to demonstrate a checkpoint, is missing to date. Taken together

there is a need to identify more of the factors involved in parasite proliferation and generate additional highly resolved, quantitative cell biological data.

A key effector of nuclear division during schizogony is the centriolar plaque (CP), the centrosome of *P. falciparum*. In mammals, centrosomes act as microtubule organizing centers (MTOC) and need to duplicate only once per cell cycle [28]. The CP of asexual blood stage parasites multiplies once each nuclear cycle and contains very few conserved centrosomal proteins [29–31]. The CP serves as a nucleation site for nuclear microtubules, while no cytoplasmic microtubule species, similar to interphase microtubules in vertebrate cells or astral microtubules in budding yeast, are observed in early schizonts [3,32–35]. Prior to the first nuclear division a monopolar spindle, also called hemispindle, with 2–11 long microtubules radiating from the intranuclear compartment of the CP, is formed [35,36] (Fig 1A). Shortly thereafter centrin, the only currently known extranuclear CP marker protein, is recruited [35,37,38]. Centrins are a group of calcium binding proteins that are associated with virtually all eukaryotic centrosomes and have been implicated in their duplication [29,39,40]. A key interaction partner of centrin in most organisms, including the apicomplexan Toxoplasma gondii, is Sfi1 and the presence of an orthologue has been hypothesized for *P. falciparum* [30,41–44]. Shortly after centrin recruitment the hemispindle collapses, the centrin signal duplicates and the mitotic spindle assembles [35]. The first nuclear division is concluded by segregation of replicated chromosome sets by an extending spindle and karyofission [13]. Subsequent asynchronous nuclear division cycles lead to a multinucleated schizont (Fig 1A).

How intranuclear microtubule dynamics are regulated in malaria parasites is unclear, but several microtubule motor proteins have been described [45,46]. Tubulin polymerization into microtubules is a concentration-dependent effect [47–49], which led us to hypothesize that hemispindle formation is an effect of high nuclear tubulin concentration [35]. How nuclear tubulin homeostasis is regulated is currently unknown.

In this study, we identify a novel centrin-interacting protein in the outer centriolar plaque and possible orthologue of Sfi1. Its knock down causes an increase in intranuclear microtubules, implicating the outer CP in regulation of tubulin homeostasis. Using time-lapse microscopy, we reveal important delays in the elongation of the aberrant spindles.

## Results

### Sfi1-like protein interacts with centrin at the centriolar plaque

To identify novel components of the centriolar plaque we carried out co-immunoprecipitation on a *P. falciparum* 3D7 strain episomally expressing PfCen1-GFP using a GFP-antibody without prior protein crosslinking (S1 Fig). Mass spectrometry analysis of the eluate revealed 11 proteins (S1 Table), of which 8 were previously identified in GFP control pulldowns using the same protocol [50]. This left three hits that were specific to PfCen1-GFP immunoprecipitation. Beside PfCen1 itself and PfCen3, we found a protein of unknown function, PF3D7_0710000. The interaction between centrins is consistent with previous findings in *P. berghei* [38]. PF3D7_0710000 is a weakly expressed and highly essential protein [51], with a size of 407kDa and was previously hypothesized to be a functional orthologue of Sfi1 [30]. This prediction was based on the presence of multiple centrin-binding consensus motifs. Our own sequence analysis using a minimal centrin-binding motif, [FLW]XXW[KR] [52], indeed shows the accumulation of those motifs within Sfi1 proteins of divergent species (Fig 1B) and revealed that PF3D7_0710000 has the highest number of centrin binding motifs of all *P. falciparum* proteins. Therefore, we decided, despite the absence of sequence homology, to name PF3D7_0710000 the Sfi1-like protein (PfSlp). To localize PfSlp, we integrated an endogenous C-terminal GFP tag using selection-linked integration [53] (Fig 1C). Additionally, we added a

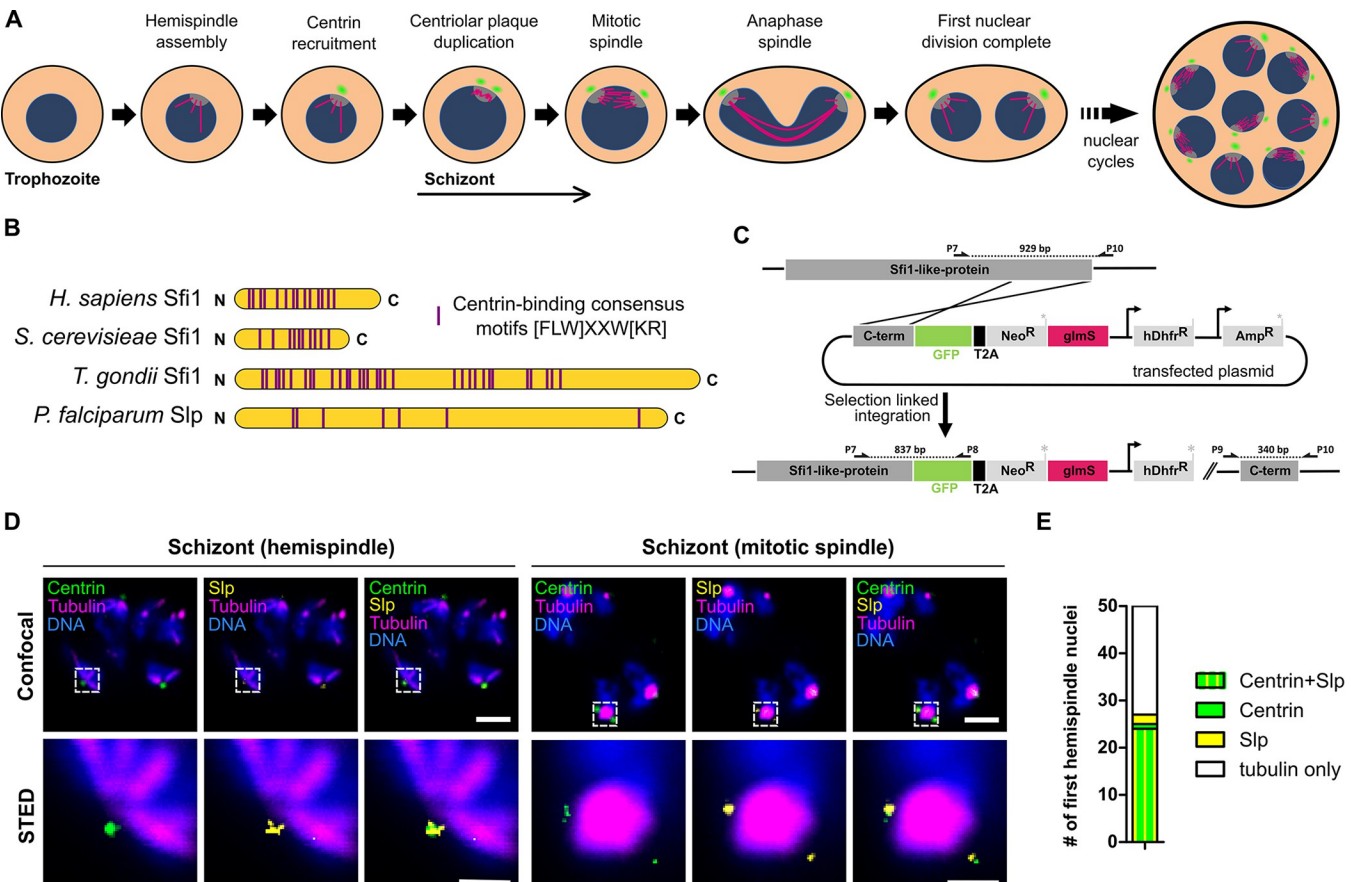

**Fig 1. The Sfi1-like protein (PfSlp) co-localizes with centrin at the centriolar plaque.** A) Schematic of first nuclear division in Plasmodium falciparum during blood stage schizogony (red blood cell not shown). After polymerization of hemispindle microtubules (magenta) at the inner core of the CP (grey), centrin (green) is recruited to the outer core of the CP. The centriolar plaque is duplicated, and the mitotic spindle is formed. During anaphase chromosomes (blue) are segregated and two nuclei are formed. Subsequent asynchronous nuclear division cycles lead to a multinucleated cell stage, which later gives rise to multiple daughter cells. B) Schematic comparison of human, Saccharomyces cerevisiae, and Toxoplasma gondii Sfi1 with PfSlp. No clear sequence homology to other Sfi1 proteins exists, but multiple centrin-binding site consensus motifs can be identified. C) Endogenous PfSlp tagging strategy with GFP and glmS ribozyme via selection-linked integration (SLI). D) Confocal microscopy images of immunofluorescence staining of blood stage schizont expressing endogenously tagged PfSlp-GFP using anti-PfCen3, which labels multiple centrins (green), anti-tubulin (magenta), anti-GFP (yellow) antibodies, and DNA stained with Hoechst (blue). Dual-color STED microscopy of marked sub-region (dotted rectangle) with super-resolved PfSlp-GFP and centrin signal. Maximum intensity projections are shown for confocal images. Scale bars, confocal, 1.5 μm; STED, 0.5 μm. E) Quantitative analysis of PfSlp, centrin and tubulin immunostained parasites in mononucleated schizonts with hemispindle. CPs were scored for presence (27/50) and absence (23/50) of centrin and/or PfSlp signal.

glmS ribozyme sequence to enable glucosamine (GlcN)-mediated knock down of *PfSlp* mRNA [54]. Correct integration was validated by PCR (S2 Fig). The resulting 3D7_pSLI_PfSlp-GFP_*glmS* parasite line will hereafter be referred to as 'Slp'. Due to its low expression PfSlp-GFP was not directly detectable in live cells. Immunofluorescence staining with an anti-GFP antibody, however, revealed a specific signal at the spindle poles (Fig 1D). As outer centriolar plaque marker we used our polyclonal anti-PfCen3 antibody [35], which based on the high sequence identity between centrins might retain some affinity for PfCen1, 2 and 4. Recent data also suggests that all four centrins interact and colocalize at the outer centriolar plaque [38,55]. Increased spatial resolution of STED microscopy still showed a PfSlp signal proximal to centrin (Fig 1D). To confirm their interaction, we carried out a reciprocal IP using anti-GFP antibody on Slp and the wild type strain as control and identified a specific centrin band in the western blot analysis (S3 Fig). Ring and trophozoite stages did not show any PfSlp signal

(S4 Fig). Upon transition into the schizont stage, late trophozoites develop a hemispindle in their nucleus of which about half carry a centrin signal [35] (Fig 1A). Our co-staining of PfSlp in those stages showed that 24 out of 25 centrin-positive centriolar plaques also had a PfSlp signal, while 24 out of 26 PfSlp-positive centriolar plaques also showed centrin (Fig 1E) indicating that both proteins are recruited simultaneously.

## PfSlp is required for blood stage parasite multiplication

To investigate the function of this novel centriolar plaque component we induced conditional self-cleavage of PfSlp-GFP_*glmS* mRNA by treatment of rings with 3.5 mM GlcN. After 73 h we measured a 55% reduction in steady state PfSlp mRNA levels by qPCR (Fig 2A). This could be recapitulated by a significantly reduced PfSlp-GFP signal at the centriolar plaque measured at the centriolar plaques of immunostained schizonts (S5 Fig). Endogenous centrin levels at the centriolar plaque were also reduced (Fig 2B), suggesting that centrin recruitment might depend on PfSlp. Slp knock down further caused a growth defect as evidenced by a reduced multiplication rate following the first cycle (24–72 hpi) after GlcN treatment (Fig 2C) and a slowed increase in parasitemia (S6 Fig). Although GlcN also influenced growth of 3D7 wild type (wt), as described previously [56], the growth defect was markedly stronger in the Slp line and still significantly different from 3D7 when correcting for GlcN treatment and PfSlp tagging effects (Fig 2C). To test whether DNA replication was altered we analysed RNase-treated, SYBR-stained cultures by flow cytometry while progressing through the first schizont stage after GlcN addition (Fig 2D). GlcN treatment caused a delay in initiation of DNA replication as indicated by curve showing cells with >2N DNA content being shifted. DNA content in Slp +GlcN parasites did not strongly differ from 3D7 +GlcN, although a small dip at 30 hpi could indicate that some parasites are dying. Slp -GlcN parasites did not reach the same peak DNA content levels, but the differences to 3D7 -GlcN were not significant. Measurement of mean DNA fluorescence intensity corroborates these observations (S7A Fig) and quantification of the area under the curve as proxy for total DNA replication showed no significant difference between any conditions (S7B Fig). The different peak height could result from 3D7 -GlcN cells remaining more synchronous. To analyze final merozoite number separately from the

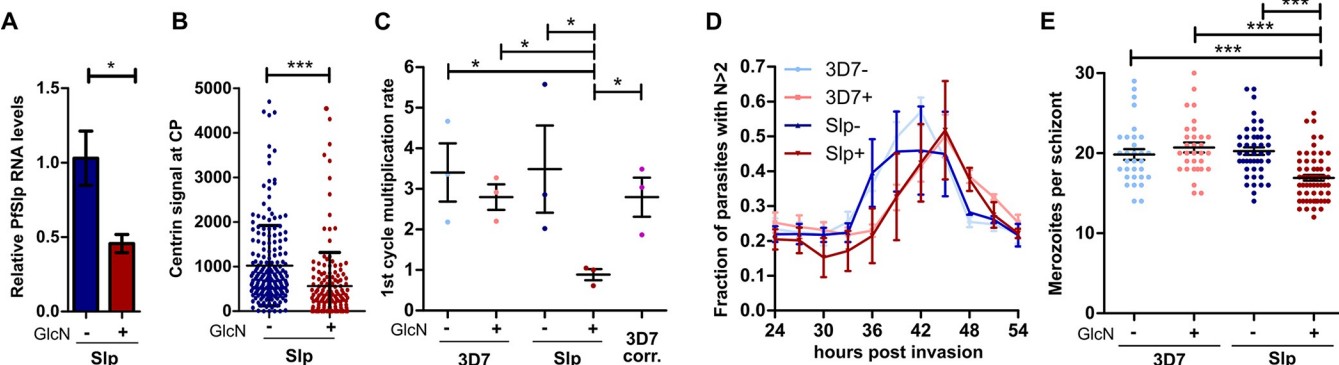

**Fig 2. PfSlp knock down impedes parasite growth and nuclear multiplication.** A) Real time qPCR analysis of PfSlp mRNA levels in Slp parasite line treated with 3.5 mM GlcN for 73h (+) or left untreated (-). Serine-tRNA ligase was used as control gene. B) Relative fluorescence signal intensity of centrin signal at the centriolar plaque was measured in +/-GlcN Slp schizont parasites with up to 10 nuclei immunostained as in Fig 1D. C) First cycle after treatment multiplication rate (24-72hpi) of 3D7 wild type and Slp strain +/- GlcN. Parasitemia was measured via flow cytometry after SYBR-Green staining and fixation. "3D7 corr." indicates the 3D7 value corrected for Slp-tagging and +GlcN effect. D) Flow cytometry analysis of SYBR-Green stained and RNase treated 3D7 and Slp parasites +/-GlcN showing fraction of parasites with >2N DNA content. E) Giemsa-based merozoite counting of late stages schizonts +/-GlcN. Egress inhibitor E64 was added at 50μM for 4h at the end of schizogony before analysis of segmenters. 3D7 +/-GlcN (n = 31/31), Slp +/-GlcN (n = 62/45). All means generated from three biological replicates (N = 3). SEM is shown. Statistical analysis by t-test with Welch´s correction. *: p<0.05, ***: p<0.0001 ns: p>0.05.

confounding factors of bulk population averages we enriched for segmenter stages and found a reduction of daughter cells specifically in GlcN-treated Slp parasites suggesting a defect in nuclear division (Fig 2E). This reduced merozoite number might not be reflected in the DNA content analysis, due to the higher variability of the flow cytometry assay or some of the Slp +GlcN merozoites being polyploid due to failed chromosome segregation.

## PfSlp knock down causes increased nuclear tubulin levels

Since the defect in nuclear multiplication was specific to PfSlp knock down rather than GlcN treatment (Fig 2E) we analysed the microtubule organization in Slp parasites, to which GlcN was added during the preceding ring stage, by immunofluorescence staining (Fig 3). Nuclei with hemispindles contained more prominent microtubule branches in GlcN-treated cells (Fig 3A). Indeed, quantification of the cumulative microtubule length per hemispindle in schizonts revealed a significant increase upon PfSlp knock down (Fig 3B). Individual microtubules cannot be distinguished once nuclei form mitotic spindles, but the overall tubulin signal intensity was high, and we observed clear microtubule protrusions for 18 out of 40 (45%) spindles in GlcN treated parasites, while only 6 out of 42 (14%) mitotic spindles in control cells deviated from the ovoid shape (Fig 3C). Since the nucleoplasmic tubulin signal intensity was not above the cytoplasmic or extracellular background, we assume that the majority of intranuclear tubulin is in the polymerized form. We therefore quantified the spindle-associated tubulin signal and found significantly higher abundance in GlcN-treated parasites for hemispindles and mitotic spindles, indicating a restrictive role of PfSlp in nuclear tubulin homeostasis (Fig 3D). To investigate the cause of increased intranuclear tubulin we quantified the cellular tubulin pool by western blot analysis of total protein extract of Slp parasites at 24, 30, and 36 hpi +/-GlcN (S8A Fig). Quantification of tubulin protein band intensity, normalized to differences in aldolase levels (S8B Fig), revealed that total tubulin levels were not significantly different in GlcN-treated parasite population while entering schizont stage around 24–36 hpi (Fig 3E). Taken together this renders the increase in nuclear tubulin through globally higher tubulin expression unlikely. Additionally, we observed no significant differences in centrin protein

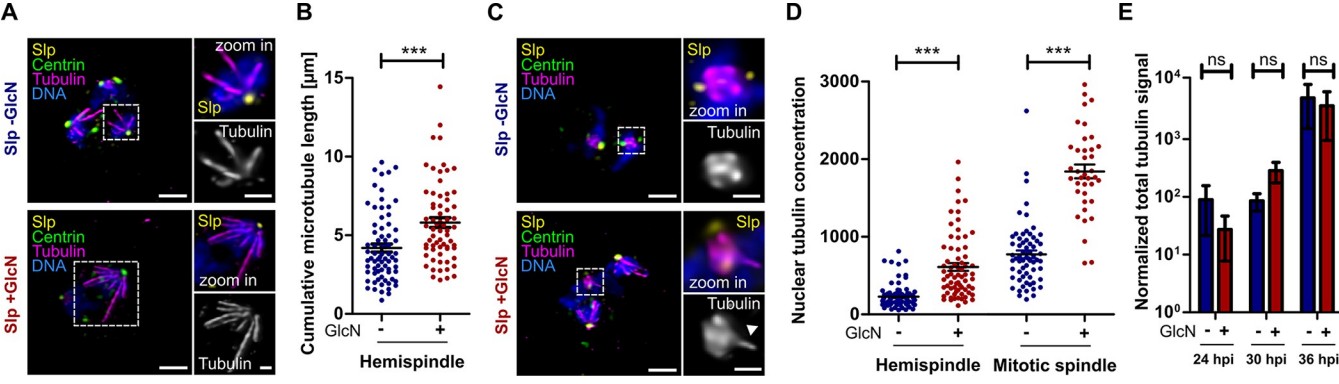

**Fig 3. PfSlp knock down causes increase of intranuclear microtubule mass.** A) Lightning deconvolution confocal microscopy image of Slp strain schizont stained with anti-centrin (green), anti-tubulin (magenta), and anti-GFP (yellow) antibody. DNA stained with Hoechst (blue). Maximum intensity projections are shown. Hemispindles show extended microtubules in +GlcN condition (bottom) Scale: 1.5 μm, zoom ins: 0.5 μm. B) Quantification of cumulative microtubule branch length in Slp schizont parasites reveal a significant increase in +GlcN (n = 66) vs -GlcN (n = 73) parasites. C) Staining and image acquisition as in A) reveals aberrant mitotic spindles in Slp +GlcN schizonts that display protruding microtubule branches (bottom). D) Nuclear tubulin signal concentration was determined on images as seen in A) and C) using polygon selections for individual hemispindles or mitotic spindles in average intensity projections after nuclear and extracellular background subtraction. Hemispindles (-GlcN (n = 91), +GlcN (n = 70)), mitotic spindles (-GlcN (n = 65), +GlcN (n = 38)). E) Western blot analysis of whole cell lysate of synchronized Slp strain at 24, 30, and 36 hpi +/- GlcN using anti-tubulin antibody normalized to difference in aldolase levels and loaded parasites. Plotted on log scale. All means generated from three biological replicates (N = 3). SEM is shown. Statistical analysis by t-test with Welch´s correction. ***: p<0.0001, ns: p>0.05.

band intensity between +/- GlcN (S8C Fig), which could account for the different centrin levels observed at the centriolar plaque in PfSlp knock down conditions using immunofluorescence (Fig 2B).

## PfSlp knock down prevents timely spindle extension

To quantify the effects of excessive nuclear tubulin abundance and errors in the mitotic spindle organization on division we used time-lapse imaging of parasites entering schizont stage stained with a live-cell compatible microtubule dye, SPY555-Tubulin. In Slp -GlcN conditions 95% (n = 18) of parasites successfully completed their first mitotic division, as indicated by full spindle elongation and duplication of the spindle signal (Fig 4A and S1 Movie). Since replication suffered a slight GlcN-dependent delay we also analysed 3D7 +GlcN where also 95% (n = 20) of cells extended their spindles (Fig 4A and S2 Movie). Strikingly, in Slp +GlcN knock down parasites, only 53% (n = 36) completed the first division, while the 47% of parasites, in which presumably the knock down was most effective, failed to do so often after several short spindle extensions and contractions (Fig 4A and S3 Movie). For those cells which underwent spindle extension we quantified the duration of the mitotic spindle phase, i.e. time from mitotic spindle formation until first notable extension, and observed a significant prolongation to 307 min on average in Slp +GlcN cellswhile Slp -GlcN and 3D7 +GlcN cells showed a mitotic spindle phase duration of 127 and 104 min, respectively, which is consistent with previously published data [35] (Fig 4B). To test if this delay is related to a perturbed microtubule quantity, we quantified the SPY555-Tubulin fluorescence intensity of the first mitotic spindle as soon as it formed and found an increased microtubule quantity in the PfSlp knock down (S9 Fig). To test whether these defects are specific to mitosis we also quantified DNA replication over time at single-cell level using a live cell microscopy compatible DNA dye, 5-SiR-Hoechst [11,57]. We acquired the first hours of nuclear division (S4 and S5 Movies). While the multiplication of nuclei in Slp KD parasites was reduced the increase in DNA signal was indistinguishable from Slp -GlcN parasites (Fig 4C). This corroborates our flow cytometry data (Fig 2) and indicates that specifically mitotic spindle function is affected. Taken together this study characterizes a novel possible Sfi1 orthologue, PfSlp, which interacts with centrin and plays a role in the homeostasis of nuclear tubulin during schizogony (Fig 4D).

## Discussion

Our PfCen1-GFP co-immunoprecipitation identified a very short list of hits, of which only PfSlp and PfCen3 were specific when compared to a GFP control. Interaction between different centrins has been previously documented in *Plasmodium* [38]. The omission of cross-linking and PfSlp expression levels being relatively low therefore indicates a direct interaction that is corroborated by reciprocal IP and previously published yeast two-hybrid data [58]. Our analysis does, however, due to the lack of quantitative enrichment analysis, not represent an exhaustive analysis of PfCen1 interaction partners. This would require combining multiple replicas of different centrin immunoprecipitation approaches with directly integrated controls and label-free quantification. While in confocal microscopy the PfSlp and centrin signals overlap completely we could occasionally see a small gap between signals in super-resolved STED images. In yeast, it was previously shown that the C- and N-terminus of yeast Sfi1, which has a highly elongated structure, are at a distance of 120 nm [41]. Given that the centrin binding motifs in PfSlp are more N-terminally located while the GFP-tag was placed at the C-terminus this gap can readily be explained by the large size of PfSlp.

The centrin-Sfi1 complex seems conserved throughout many eukaryotes and our findings support this for *Plasmodium* spp. [59]. In mammals Sfi1 localizes within centrioles and seems

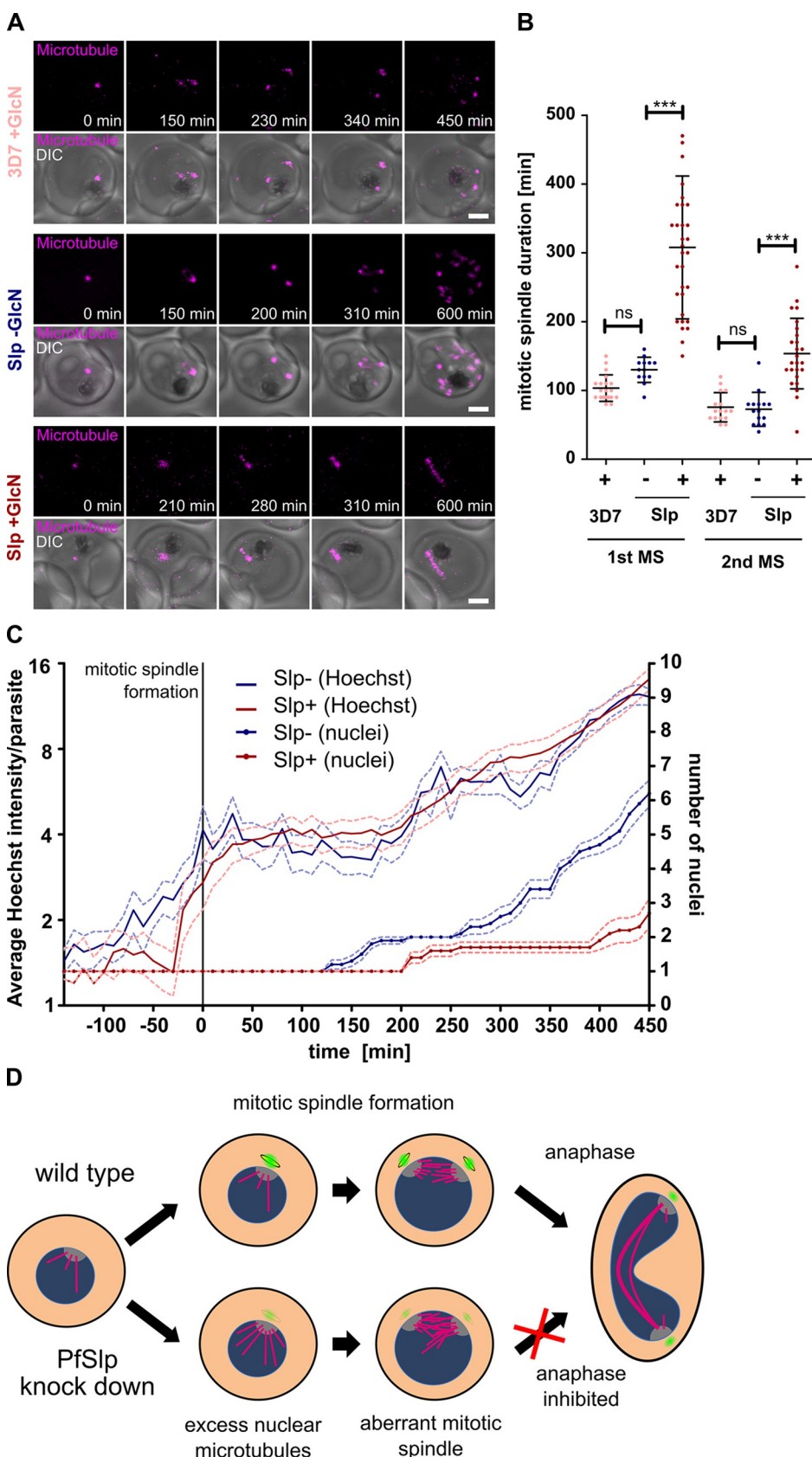

**Fig 4. Increased nuclear tubulin abundance inhibits spindle extension.** A) Time-lapse confocal imaging of 3D7 +GlcN Slp +/-GlcN parasites undergoing schizogony stained with SPY555-Tubulin (magenta). Maximum intensity projections are shown. Scale bars, 1.5 μm. B) Quantification of first (3D7+GlcN (n = 20), Slp-GlcN (n = 18), Slp+GlcN (n = 31)) and second (3D7+GlcN (n = 16), Slp-GlcN (n = 16), Slp+GlcN (n = 25)) mitotic spindle duration using live-cell movies of tubulin-stained parasites that undergo spindle extension. Statistical analysis: t-test with Welch´s correction. ***: p<0.0001, ns: p>0.05. C) Quantification of DNA content and nuclear number from time-lapse microscopy data of 5-SiR-Hoechst and SPY555-Tubulin stained Slp-GlcN (n = 10) and Slp+GlcN (n = 10) of parasites going through first mitotic spindle formation (t = 0). SEM is shown. All movies were acquired in three independent sessions (N = 3). SEM is shown. D) Schematic model of function of the centrin (green) / Slp (yellow) complex in affecting proper mitotic spindle (magenta) assembly in the nucleus (blue).

implicated in maintaining centriole architecture and cohesion [43]. Centrioles are however missing from the centriolar plaque of asexual blood stage parasites. A related structure in this context is the spindle pole body (SPB) of budding yeast, which also undergoes closed mitosis. In yeast, Sfi1 interacts with the yeast centrin homologue Cdc31 and is essential for duplication of the SPB [60]. Sfi1 is part of a structure called the half bridge, which binds Cdc31 and later rearranges into a full bridge by antiparallel binding to allow assembly of the second spindle pole [41,42,60–62]. Although we did not observe a specific defect in centriolar plaque duplication we are currently lacking independent centriolar plaque markers to investigate this question more directly. In *T. gondii*, a Sfi1 homologue was found in the outer core of the bipartite centrosome colocalizing with TgCentrin [30]. *T. gondii*, however carries centriole-like structures that could harbor the observed Sfi1 signal as evidenced by the colocalization with Sas-6, a canonical eukaryotic centriole component [29]. A temperature-sensitive TgSfi1 mutant had a loss of outer core duplication which was associated with a severe block in budding [30]. What the unifying functions of Sfi1 and centrin are in the context of these highly divergent centrosomal structures needs to be analyzed more extensively.

An impact on nuclear tubulin homeostasis has however not been described [63,64]. Higher intranuclear tubulin abundance can in principle be explained by a delay at the entry into the schizont stage caused e.g. by a failure to properly duplicate the centriolar plaque, which would then lead to an accumulation tubulin before progressing through mitosis. Since we excluded a global increase in tubulin levels in PfSlp knock down (Fig 3E), one could also hypothesize an effect on tubulin import into the nucleus. The positioning of PfSlp and centrin close to the region of the centriolar plaque which connects the extranuclear and intranuclear compartment raises the possibility that this opening in the nuclear envelope might be implicated in the import of division factors such as tubulin. This would allow the parasite to specifically regulate the nuclear levels of proteins required for DNA replication and chromosome segregation during schizogony.

Importantly, we observed that the majority of PfSlp knock down cells could not complete spindle extension while the ones that did progress were significantly delayed. This presents a very early phenotype that manifests right after recruitment of centrin and PfSlp to the centriolar plaque and is corroborated by an increase in microtubules in the first mitotic spindle as it forms (S8 Fig). This also explains the comparatively mild phenotypes at later stages, such as the only slightly decreased merozoite number (Fig 2E), as those might result from parasites for which the knock down was presumably less effective. The failure to properly progress through anaphase can be explained by the disorganization of the mitotic spindle, which relies on tightly regulated microtubule dynamics [65]. The lack of properly arranged interpolar microtubules could, e.g., reduce the pushing forces required for spindle extension. It is also plausible that proper kinetochore attachment and separation are disturbed under those conditions. If some type of spindle assembly checkpoint existed, as had been postulated in *T. gondii* [66], it could cause an anaphase delay, like the one observed in our data. The recent discovery of a Mad1

homolog in *Plasmodium* spp. further argues for the conservation of a spindle checkpoint in Apicomplexa [25]. By no means is this conclusive evidence for a mitotic checkpoint, which would among other things require the demonstration of reversibility. It, however, highlights the importance of time-resolved single-cell data, which was not available in previous studies implying the absence of a mitotic checkpoint [13,24,26,27].

Eukaryotic centrosomes remain fascinatingly diverse structures that use a mixture of common and divergent mechanisms that need to be investigated in more detail using evolutionary cell biology approaches.

## Materials and methods

### Parasite culture

The *P. falciparum* 3D7 wild type cell line and the 3D7_SLI_slp-gfp_*glmS* (Slp) strain were cultured in 0+ human red blood cells using RPMI 1640 medium (Sigma) with added 0.2 mM hypoxanthine, 25 mM HEPES, 0.5% Albumax and 12.5 µg/ml Gentamicin (cRPMI) at a hematocrit of 2.5% and an average parasitemia of 1–5%. For washing of cells, cRPMI without Albumax was used (iRPMI). For Slp, 600ug/ml Geneticin-G418 (Thermo Fisher) was added. Parasites were incubated at 90% relative humidity, 5% $O_2$ and 3% $CO_2$ at 37˚C. Parasites were synchronized by selective lysis of schizonts using 5% D-sorbitol for 10 minutes at 37˚C, performed three times throughout one week or alternatively via quadroMACS (Miltenyi Biotec) magnetic purification of late stage schizonts followed by sorbitol treatment.

### Co-immunoprecipitation

Co-immunoprecipitation was based on previous work [21,67]. Specifically, 3.75 ml of NF54 PfCentrin1-GFP schizont culture were harvested at a parasitemia of circa 5%. Therefore, red blood cells were pelleted for 5 min at 340 g and lysed with 0.1% saponin/cRPMI for 1 min at room temperature. Lysed cells were pelleted (3000 g, 6 min) and washed with 0.1% saponin/iRPMI by centrifugation (3000 g, 6 min). The following steps were performed on ice if not stated otherwise. For parasite lysis, the sample was incubated with 1 ml RIPA buffer (50 mM Tris-HCl, pH 8, 150 mM NaCl, 1% NP-40, 0.5% sodium deoxycholate, 0.1% SDS) freshly supplemented with 1 mM DTT and 1x protease inhibitor (Halt Protease Inhibitor Cocktail, 78430, Thermo Fisher Scientific) for 30 min. For mechanical homogenization, the sample was passed ten times through a needle (0.3 mm diameter) attached to a syringe. A fraction of the sample was taken as "whole cell lysate" for western blot analysis. The sample was centrifuged at maximum speed (21130 rcf) for 15 min at 4˚C. The supernatant ("input") was saved for western blot analysis. 100µl Dynabeads Protein G for Immunoprecipitation (magnetic, 10003D, Invitrogen) were washed with 1 ml 0.01% Tween-20/PBS. Beads were incubated with 3.5 µg mouse anti-GFP antibody (11814460001, Roche) for a minimum of 10 min on room temperature, rotating (S3 Table). After washing the beads twice with 1 ml 0.1% Tween-20/PBS and once with 1 ml RIPA buffer, beads were resuspended in RIPA buffer. The sample was incubated with antibody-conjugated Dynabeads over night at 4˚C, rotating. Sample was pelleted and supernatant taken as "flow-through" for western blot analysis. The bead pellet was washed 3-5x with 1 ml RIPA buffer and twice with PBS, both freshly supplemented with 1x protease inhibitor. A fraction of the bead sample ("eluate") was taken for western blot analysis. To prepare the sample for mass spectrometry analysis, PBS was exchanged by 100 µl 6 M Urea/ 50 mM ammonium bicarbonate (AB) and the sample was incubated with 2 µl of 50 mM reducing agent dithioerythritol/$dH_2O$ for 1h at 37˚C. After incubation of the sample with 2 µl of 400 mM alkylating agent iodoacetamide/ $dH_2O$ for 1h in the dark, the urea concentration was reduced to 1 M via addition of 500 µl AB. The sample was digested with 7 µl 0.1 µg/µl

trypsin (Promega) /AB over night at 37˚C. The supernatant was collected, dried completely using speed-vacuum and desalted with a C18 microspin column (Harvard Apparatus) according to the manufacturer's instructions. The sample was dried completely under speed-vacuum and stored at -20˚C.

For reciprocal Co-IP 6ml Slp and 6ml 3D7 culture were harvested at a parasitemia of circa 5%. Lysis was performed as for the first Co-IP. 60µl of the lysed cells (whole cell) as well as the supernatant (Input) and pellet (pellet) were taken for Western blot analysis. Reciprocal Co-IP was performed using GFP-Trap Magnetic Particles M-270 (ChromoTek, gtd-10), according to the manufacturers´ protocol. After incubation of input on the beads, a sample (flowthrough) was taken for western blot analysis. After washing thrice with wash buffer (10 mM Tris/Cl pH 7.5, 150 mM NaCl, 0.05% Nonidet P40 Substitute, 0.5 mM EDTA), a sample of the third wash was taken (wash 3) and the sample was eluated in 40µl 4X Laemmli buffer (1610747, Bio-Rad) and 10% beta-mercaptoethanol. By incubation at 95˚C for 5 minutes

## Mass spectrometry analysis and database search

The dried sample was dissolved in 20 µl loading buffer (5% acetonitrile, 0.1% formic acid). 2 µl of the sample were injected on column. LC-ESI-MS/MS was performed on a Q-Exactive HF Hybrid Quadrupole-Orbitrap Mass Spectrometer (Thermo Scientific) equipped with an Easy nLC 1000 liquid chromatography system (Thermo Scientific). Peptides were trapped on an Acclaim pepmap100, C18, 3µm, 75µm x 20mm nano trap-column (Thermo Scientific) and separated on a 75 µm x 250 mm, C18, 2µm, 100 Å Easy-Spray column (Thermo Scientific). For analytical separation, a gradient of 0.1% formic acid/$H_2O$ (solvent A) and 0.1% formic acid/ acetonitrile (solvent B) was run for 90 min at a flow rate 250 nl/min. The following gradient was used: 95% A/5% B for 0–5 min, 65% A/35% B for 60 min, 10% A/90% B for 10 min and finally 10% A/90% B for 15 min. The MS1 full scan resolution was set to 60'000 full width half maximum (FWHM) at a mass/charge (m/z) ratio of 200 with an automatic gain control (AGC) target of 3 x $10^6$ (number of ions), a maximum injection time of 60 ms and a mass range set to 400–2000 m/z. For data dependent analysis, up to twenty precursor ions were isolated and fragmented by higher-energy collisional dissociation at 27% normalized collision energy. MS2 scans were acquired in centroid mode with the resolution set to a FWHM of 15'000 at m/z 200 with an AGC target of 1 x $10^5$ and a maximum injection time of 60 ms. Isolation width was set to 1.6 m/z. Full MS scans were acquired in profile mode. The dynamic exclusion was set to 20 s.

A peak list was generated from the raw data by using the conversion tool MS Convert (ProteoWizard). Files of the peaklist were searched against the Plasmo DB 3D7 database (release 46, 5548 entries) and combined with an in-house database containing common contaminants using Mascot (Matrix Science, version 2.5.1). For the Mascot search, trypsin was selected as the enzyme with one potential missed cleavage and tolerances of precursor ions and fragment ions were set to 10 ppm and 0.02 Da, respectively. Carbamidomethyl cysteine was set as fixed amino acid modification. Variable amino acid modifications were oxidized methionine, deaminated and phosphorylated serine, threonine, and tyrosine. For validation of the Mascot search, Scaffold 4.10.0 (Proteome Software) was used. The identified peptides were accepted when the probability to reach an FDR of less than 0.1% was higher than 44% with the Peptide Prophet algorithm [68] with Scaffold delta-mass correction. Protein identification was accepted when the probability was greater than 99% to reach a false discovery rate (FDR) of less than 1% and when at least two peptides were identified. Protein probabilities were assigned by the Protein Prophet algorithm [69]. Proteins with similar peptides that could not be differentiated based on MS/MS analysis alone were grouped to satisfy the principles of parsimony.

## SDS page and Western blotting

Samples for western blot analysis taken during co-immunoprecipitation ("whole cell lysate", "flow-through", "input" and "eluate") were each supplemented with NuPAGE LDS Sample Buffer (4X) (NP0007, Invitrogen), 50 mM DTT and 5% beta-mercaptoethanol. After sample incubation for 5 min at 95˚C, beads in the "eluate" sample were removed via a magnetic rack and only supernatant was used. Per lane, protein lysate of $1 \times 10^7$ cells was loaded on a 4–12% tris-glycine gel (Novex WedgeWell, XP04122BOX, Thermo Fisher Scientific) and run in tris-glycine buffer (25 mM Tris, 250 mM glycine, 0.1% SDS in $H_2O$). Proteins were transferred to a nitrocellulose membrane using the Mini Trans-Blot Cell (1703930, Bio-Rad) system by blotting in tris-glycine based buffer (25 mM Tris, 192 mM glycine, 0.02% SDS, 0–25% methanol in deionized water), for 14 h at 40 mA on ice. All the following incubation steps of the membrane were performed at room temperature with slight agitation. First, the membrane was briefly rinsed in deionized water and incubated with Ponceau solution (0.1% (w/v) in 5% acetic acid, Ponceau S solution, P7170, Sigma-Aldrich) for 3–5 min. Second, it was briefly rinsed in deionized water and immediately imaged using the ChemiDoc XRS+ Gel Imaging System (Bio-Rad) equipped with the Image Lab 4.1 software. Next, the membrane was briefly incubated in deionized water and washed with 0.05% Tween-20/PBS. Nonspecific binding sites were blocked with 5% milk powder/0.05% Tween-20/PBS for 30 min, exchanged again for blocking solution and incubated for another 15 min. The membrane was stained with primary antibody mouse anti-GFP in 5% milk powder/0.05% Tween-20/PBS for 1 h (S3 Table). After three washes with 0.05% Tween-20/PBS for 10 min, the membrane was stained with the secondary antibody goat anti-mouse IgGHRP in 5% milk powder/0.05% Tween-20/PBS for 1h. After 3x 10 min washing steps with 0.05% Tween-20/PBS, the membrane was incubated with 1:1 Amersham ECL Prime Western Blotting Detection Reagent (RPN2232, Amersham) for 5 min and immediately imaged using the ChemiDoc system.

Samples for western blot analysis of tubulin and centrin abundance were taken of +/- 2.5h synchronized Slp parasites with or without 3.5mM GlcN after 24, 30 or 36 hpi by lysis in 0.15% saponin in PBS, followed by supplementation in 4X Laemmli buffer (1610747, Bio-Rad) and 2.5 beta-mercaptoethanol. Samples were incubated for 10 min at 95˚C and lysate of $2*10^7$, $1*10^7$ and $2*10^6$ cells was loaded for 24, 30 and 36 hpi respectively on a 4–15% Mini-PRO-TEAN TGX Precast Protein Gel (4561083, Bio-Rad). Blotting was performed using the Trans-Blot Turbo Mini 0.2 µm Nitrocellulose Transfer Pack (1704158, Bio-Rad) on a Trans-Blot Turbo Transfer System (Bio-Rad). Thereafter, the membrane was blocked for 15 min in 5% milk powder in 0.5% Tween-20/PBS (5% milk). The blot was cut at the 25kDa band, and the upper part was incubated with rabbit anti-PfAldolase and mouse anti-tubulin (S3 Table), while the lower part was incubated with rabbit anti-Centrin3. Antibodies were always diluted in 5% milk, and staining occurred for 2h at Rt while shaking. The membrane was washed thrice for 10 min in 5% milk and incubated with anti-mouse 800CW and anti-rabbit 680R (upper part), or anti-rabbit 800CW (lower part) antibodies for 1h at RT shaking (S3 Table). The blot was washed thrice in 5% milk for 10 min and once in PBS for 10 min before imaging on the LI-COR Odyssey DLx Imaging System.

## Generation of the pSLI_PfSlp-GFP-*glmS* plasmid

To generate the pSLI_PfSlp-GFP-*glmS* plasmid, pSLI-CRK4-GFP-*glmS* plasmid (kindly provided by Markus Ganter) was digested with NotI-HF and MluI-HF to remove the CRK4 sequence. PfSlp genomic sequence was amplified via PCR from NF54 genomic DNA (forward primer: TGACACTATAGAATACTCGCGGCCGCAGTGAAAAGACTGATGAAGG, reverse primer: CAGCAGCAGCACCTCTAGCACGCGTTTTTATCATGATAAGATTGTTAAGG).

Ligation of backbone and PfSlp gene sequence was performed using Gibson assembly [70]. To verify the plasmid sequence, an analytical digest was performed followed by Sanger sequencing.

## Parasite transfection

For transfection of 3D7 parasites with pSLI_PfSlp-GFP-*glmS*, 150 μl packed red blood cells infected with sorbitol-synchronized ring stages (~4% parasitemia) were mixed with 100 μg purified DNA in TE buffer and 300 μl prewarmed Cytomix (120 mM KCl, 0.15 mM CaCl2, 2 mM EGTA, 5 mM MgCl2, 10 mM K2HPO4/KH2PO4 pH 7.6, 25 mM HEPES pH 7.6). Parasite transfection was performed via electroporation using Gene Pulser II (Bio-Rad) at high capacity, 310 V and 950 μF. To select for transfected parasites, 2.5 nM WR99210 (Jacobus Pharmaceuticals) were added to the culture. Selection for construct integration was performed according to the protocol by Birnbaum et al., 2017 [53] using 800 μg/ml Geneticin-G418 (Thermo Fisher Scientific). To check for correct integration, genomic DNA was extracted, and PCRs performed across the 5' and 3' integration site (S2 Fig). Primers used are found in S2 Table.

## Seeding of infected erythrocytes

Treatment with 3.5 mM GlcN or control treatment always occurred during the ring stage preceding the schizont stage targeted for analysis. For IFA, infected RBCs were seeded on eight-well chambered glass slides (glass bottom μ-Slide 8 Well, ibidi) by first coating the glass surface with Concanavalin A (Sigma-Aldrich, 5 mg/ml in ddH$_2$O) for 20 min at 37˚C. For live-cell imaging, precoated 8 well slides (ibiTreat μ-Slide 8 Well, ibidi) were washed thrice briefly with prewarmed Dulbecco´s phosphate-buffered solution (PBS) (Gibco). Thereafter, the slides were rinsed thrice with prewarmed incomplete iRPMI 1640 medium. 150μl of infected erythrocyte culture per well was washed twice with incomplete RPMI and added to the coated slides. After letting the cells bind for at least 10 min at 37˚C unbound cells were removed via repeatedly shaking the slide and washing with 200μl iRPMI, until only a cell monolayer remained. Finally, cells were either incubated in 200μl complete medium at 37˚C until live-cell imaging setup, or briefly rinsed with PBS before fixation in 4%PFA/PBS for 20 min at 37˚C. Fixed cells were rinsed thrice with PBS before storage at 4˚C or direct use for IFA.

## Immunofluorescence assay

For IFA, seeded and PFA-fixed parasites were first permeabilized with 0.1%Triton-X-100/PBS for 15 minutes at RT and rinsed thrice with PBS. Afterwards, free aldehyde groups were quenched by incubation in fresh 0.1mg/ml NaBH$_4$ solution at RT for 10 min. After brief three-time rinsing, blocking was performed in 3% Bovine Serum Albumine (BSA) (Carl Roth) in PBS for 30 minutes. Primary antibodies targeting GFP, PfCentrin3 and tubulin were diluted to the according concentration in 3% BSA and incubated on the cells for 2h at RT while shaking (S3 Table). Primary antibodies were thereafter removed, and the dish was washed thrice with PBS-T for 10 min shaking before incubation with secondary antibodies and Hoechst (S3 Table) for 90 min at RT. Cells were washed twice in PBS-T for 10 min, once in PBS and stored in PBS at 4˚C until imaging.

## Staining of infected red blood cells for live-cell imaging

For live cell imaging, Slp or 3D7 wt infected red blood cells (iRBCs) were seeded on dishes as described above. Cells were synchronized with sorbitol and treated with 3.5mM GlcN 24h

before imaging. Imaging medium consisting of phenol red-free RPMI 1640 supplemented with L-Glutamine as well as the same supplements used in complete medium for culture was equilibrated in the cell culture incubator overnight. The live microtubule dye SPY555-Tubulin (Spirochrome) (S3 Table) and 3.5mM GlcN were added into the imaging medium, and thereafter used for replacement of the culture medium of the seeded cells. For DNA quantification 20 nM 5-SiR-Hoechst was added. The dish was closed airtight without residual air and incubated for 2h prior to imaging.

## Confocal and STED microscopy

Confocal microscopy of fixed cells was performed on a Leica TCS SP8 scanning confocal microscope (Leica) with Lightning (LNG) automated adaptive deconvolution. Images were acquired using the HC PL APO CS2 63x/1.4 N.A. oil immersion objective, HyD detectors, and spectral emission filters. Brightfield images were obtained from a transmitted light PMT detector. In LNG mode, images were acquired using a pinhole of 0.5 airy units resulting in a pixel size of 29 nm and a total image size of $9.26 \times 9.2$ µm ($320 \times 320$ pixels) with a pixel dwell time of 2.3µs. Z stacks were acquired at a total size of 6.27µm at a z-step size of 230 nm. STED microscopy was performed on a single-point scanning STED/RESOLFT super-resolution microscope (Abberior Instruments GmbH), equipped with a pulsed 775 nm STED depletion laser and three avalanche photodiodes for detection. Super-resolution images were acquired using the 100× 1.4 NA objective, a pixel size of 20 nm and a pixel dwell time of 10 µs. The STED laser power was set to 5–15%, whereas the other lasers (405, 488, 594 and 640) were adjusted to the antibody combinations used (S3 Table).

## Live-cell imaging

Live-cell imaging was performed on a Zeiss LSM 900 microscope (Zeiss) equipped with the Airyscan detector using Plan-Apochromat 63x/1,4 oil immersion objective at 37˚C in a humidified environment. Movies were acquired at multiple positions using an automated stage and the Definite Focus module for focus stabilization with a time-resolution of 10 min for 18–21 h. Images were acquired sequentially in the sequence scanning mode using 567 nm diode lasers for SPY555-Tubulin imaging. Emission detection was configured using variable dichroic mirrors. Detectors were used with the gain adjusted between 700 and 900 V, offset was not adjusted (0%). Pinhole was set to 0.6 Airy Units. Brightfield images were obtained from a transmitted light PMT detector. Sampling was sized 19.11 µm x 19.11 µm at 50 nm pixel size xy bidirectionally with pixel dwell time between 0.7 and 1.2 µs without line averaging. Z-stacks of a total size of 6 µm at a z-step size of 350 nm were acquired. Subsequently, ZEN Blue 3.1 software was used for the 3D Airyscan processing with automatically determined default Airyscan Filtering (AF) strength. Trophozoites were identified in DIC and presence of a hemispindle was determined prior to imaging.

## Image analysis and quantification

All graphs were plotted using Prism GraphPad. Statistical significance was determined via unpaired t-test with Welch´s correction. All images were analyzed using ImageJ [71]. If not stated otherwise, mean and SEM are plotted.

**Representative images.** Confocal images shown are maximum projections, brightness and contrast were adjusted in the same manner for +/- GlcN samples for visualization. Movie stills were acquired from maximum intensity projections with adjusted tubulin signal. Brightness and contrast of STED images were adjusted accordingly.

**Centriolar plaque signal quantification.** To determine the centrin and PfSlp intensity, we specifically selected parasites with 3–10 nuclei and full z-stacks of about 6 μm including the entire tubulin, centrin, PfSlp and DNA stained PfSlp parasites + (n = 31) or—(n = 28) GlcN were acquired and projected. Intensity of each centriolar plaque signal was then measured using a circle with an area of 0.14 μm$^2$ in ImageJ around the centrin/PfSlp signal. Extracellular and intracellular background was subtracted. Negative values were recorded as "0". Determined signal (+GlcN (n = 140), -GlcN (n = 195) was plotted thereafter. Mean and SD shown.

**Microtubule length determination.** To calculate the average cumulative length of microtubule strands in hemispindles per nucleus, average intensity projections of tubulin, centrin, PfSlp and DNA stained Slp schizonts +GlcN (n = 31) or -GlcN (n = 28) were acquired. Length of individual microtubule strands was then measured using the line tool in ImageJ. Length of each microtubule strand was cumulated for every hemispindle (+GlcN (n = 66), -GlcN (n = 73)).

**Nuclear tubulin concentration.** To calculate the average nuclear tubulin concentration in hemispindles and mitotic spindles, average intensity projections of tubulin, centrin, GFP stained Slp schizonts + (n = 31) or—(n = 28) GlcN were acquired. Mean intensity of each spindle was then measured using the polygon tool in ImageJ drawn around the tubulin signal. Intensity was divided by measured area to determine concentration after background subtraction. Concentration of hemispindles (+GlcN (n = 70), -GlcN (n = 89)) and mitotic spindles (+GlcN (n = 38), -GlcN (n = 64) was determined.

**Western blot quantification.** Protein band signal intensity was determined using Image Studio. Centrin and tubulin levels were normalized to equal aldolase signal and corrected to display 2*10$^7$ parasites.

**Parasite spindle duration.** To measure the duration of the first mitotic spindle, Slp (n = 31) and 3D7 (n = 20) parasites were previously treated with 3.5mM GlcN and movies were created as described above. Mitotic spindle duration was defined as first frame with identifiable mitotic spindle up to the last frame with identifiable mitotic spindle.

**Live cell microscopy microtubule concentration.** To measure the duration of the first mitotic spindle, movies of Slp (n = 29) and 3D7 (n = 19) parasites previously treated with 3.5mM GlcN or untreated Slp parasites were created as described above. Average intensity projections of z-slides containing the mitotic spindle at the first time frame with identifiable mitotic spindle. Microtubule intensity was then measured using a circle tool in ImageJ drawn around the mitotic spindle. Intracellular background was subtracted. Mean and SD are shown.

**Percentage of failed anaphase spindles.** Relative failure to extend the anaphase spindle was determined by creation of live cell data as described above. Failure was defined as multiple extension attempts without severing of the anaphase microtubule strands. Slp +GlcN (n = 36), Slp -GlcN (18), 3D7 +GlcN (n = 20).

**Single cell time lapse DNA quantification.** To quantify total cellular DNA content in time lapse movies of 5-SiR-Hoechst and SPY555-Tubulin stained Slp-GlcN (n = 10) and Slp +GlcN (n = 10) we generated average intensity projections from z-stacks containing the total cell. We selected a sufficiently large and consistent region of interest and quantified mean DNA signal for each time point followed by background subtraction. All data sets were aligned to the timepoint of mitotic spindle formation (i.e. collapse of hemispindle) based on the tubulin signal. Nuclear number was counted in parallel using 5-SiR-Hoechst.

## Parasite growth assay

PfSlp and 3D7 parasites synchronized via quadroMACS (Miltenyi Biotec) magnetic purification of late stage schizonts followed by sorbitol treatment after 4h of reinvasion were treated

with +/- 3.5mM GlcN and seeded with a parasitemia of 0.1%. Growth assay was performed as described previously [19]. 100µl resuspended medium were taken and fixed with 4% PFA/PBS and 0.0075% Glutaraldehyde for 30 min at RT every 24 hpi. Samples were washed, permeabilized with Triton X–100 (Sigma, T8787) for 8 min and treated with 0.3mg/ml Ribonuclease A (Sigma, R4642) in PBS for 30 min. Cells were washed twice and resuspended in 100µl staining solution (1:2000 SYBR green I (Invitrogen) in PBS) followed by 20 min incubation. After subsequent washing cells were diluted to the preferred dilution and parasitemia was determined via flow cytometry. Cells were gated for single red blood cells and plotted using Prism Graphpad. N = 3

## Merozoite quantification

To quantify the average number of merozoites per segmenter, sorbitol synchronized PfSlp-GFP and wt 3D7 parasites were treated with 50µM of the egress inhibiting protease inhibitor trans-epoxysuccinyl-L-leucylamido (4-guanidino) butane (E64) after 33h of treatment with Glucosamine as early ring stage parasites. Untreated cells of both strains served as control. After 4 hours E64 incubation, number of merozoites per late stage schizont was determined via giemsa smear. Statistical significance was determined using unpaired two-sided student's t-test via Prism GraphPad software.

## RT-qPCR

To determine the PfSlp RNA abundance upon knockdown, iRBCs +/- 3.5 mM GlcN were first lysed using 0.15% Saponin in PBS. Thereafter RNA was extracted using the NucleoSpin RNA kit (Macherey-Nagel, 740955.50) and cDNA was synthesized *in vitro* using the RevertAid first strand cDNA synthesis kit (Thermo Fisher, K1622). qPCR was performed using SYBR-green with primers targeting the PfSlp cDNA and t-serine ligase as housekeeping control. Knockdown was determined using the delta-delta method with averaged housekeeping genes throughout all replicas. Statistical significance was determined using Prism GraphPad. N = 3.

## Determination of S-Phase onset and DNA content via flow cytometry

For analysis of parasite DNA content via flow cytometry cells were first synchronized using quadroMACS (Miltenyi Biotec) magnetic purification of late stage schizonts followed by sorbitol treatment after 3h. iRBCs with a parasitemia of ~0.5% were thereafter treated with 3.5mM GlcN or left untreated and incubated at the beforementioned incubation conditions, after 24 hpi 200µl of medium was taken and fixed with 4%PFA/PBS and 0.0075% Glutaraldehyde for 12h at 4°C every 3h until 54 hpi. Samples were washed, permeabilized with Triton X–100 (Sigma, T8787) for 8 min and treated with 0.3mg/ml Ribonuclease A (Sigma, R4642) in PBS for 30 min. Cells were washed twice and resuspended in 100µl staining solution (1:2000 SYBR green I in PBS) followed by 20 min incubation. After subsequent washing cells were diluted to the preferred dilution and DNA content was analyzed via flow cytometry. Cells were gated for 2000 single, infected red blood cells and SYBR green emission intensity after FITC excitation was measured for each time point and condition. Intensity was normalized to equal starting emission in each individual replicate. For determination of DNA content >2, we quantified the amount of iRBCs >2N relative to the total number of iRBCs. Graphs were plotted using Prism GraphPad. N = 3.

## Supporting information

**S1 Fig. Western blot analysis shows purification of PfCentrin1-GFP proteins via co-immunoprecipitation.** "Whole-cell lysate", "input", "flow-through" and "eluate" sample fractions

were taken at different steps of PfCentrin1-GFP co-immunoprecipitation as described in the methods section. Per lane, protein lysate of $1 \times 10^7$ parasite cells was loaded (number of infected red blood cells per ml determined before Saponin-lysis during cell harvesting). Size of the bands (circa 47 kDa) detected by the anti-GFP antibody in all lanes of the western blot corresponds to the molecular weight of PfCentrin1 (19.6 kDa) tagged with GFP (26.9 kDa). In the eluate fraction some degradation of PfCen1-GFP could be observed. Please note that the weak Centrin1-GFP band in the eluate might be due to loss of protein lysate during the individual (washing) steps of the IP. This is consistent with absence of protein signal (neither for Centrin1-GFP nor for unspecific proteins) in the eluate lane by Ponceau staining.
(TIF)

**S2 Fig. PCR validation of genome integration of GFP-glmS-tag.** PCR of PfSlp-GFP transfected cells and 3D7 wildtype cells targeting 5´ and 3´ integrations to validate complete integration of the GFP-glmS tag to endogenous *PfSlp*. Control PCR targeting the unaltered locus of *PfSlp* in the 3D7 wild type strain confirms complete integration and modification of the endogenous locus.
(TIF)

**S3 Fig. Reciprocal co-immunoprecipitation confirms interaction of PfSlp with centrins.** A) Western blot analysis of co-immunoprecipitation using anti-GFP beads on 3D7 wild type whole cell lysate ('WC') on the left and Slp 'WC' on the right. Centrin was detected by anti-PfCen3 antibody. Per lane, protein lysate of $4 \times 10^7$ late stage parasite cells were loaded (number of infected red blood cells per ml determined before Saponin-lysis during cell harvesting). Other lanes contain pellet (P), supernatant (SN), flow-through (FT), third wash (W3), and eluate (E) sample fractions, which were taken at different steps of co-immunoprecipitation as described in the methods section. '-' designates empty lanes. Size of the bands in all lanes correspond to the molecular weight of centrins (~20 kDa). B) Ponceau staining shows absence of protein signal in the eluate lane.
(TIF)

**S4 Fig. PfSlp is not expressed before onset of schizont stage.** Confocal microscopy images of immunofluorescence staining of ring stage parasites and trophozoites expressing endogenously tagged PfSlp-GFP using anti-centrin, anti-tubulin and anti-GFP antibodies. DNA stained with Hoechst. Maximum intensity projections are shown. Scale bars, 1.5 µm.
(TIF)

**S5 Fig. GlcN-treatment causes reduction of PfSlp at the centriolar plaque.** Relative fluorescence signal intensity of PfSlp-GFP signal at the centriolar plaque was measured in +/-GlcN Slp schizont parasites with up to 10 nuclei, immunostained as in Fig 1D. Means generated from three biological replicates (N = 3). SEM is shown. Statistical analysis by t-test with Welch´s correction. ***: p<0.0001.
(TIF)

**S6 Fig. PfSlp knock down causes a growth defect.** Growth curves of 3D7 wild type and Slp strain +/- GlcN. Parasitemia was measured via flow cytometry after SYBR-Green staining and fixation for 5 days after treatment. Log10 of parasitemia is plotted to highlight differences in slope. All means generated from three biological replicates (N = 3). SEM is shown.
(TIF)

**S7 Fig. Onset of DNA content increase is delayed by GlcN treatment. A)** Flow cytometry analysis of SYBR-Green stained and RNase treated 3D7 and Slp parasites +/-GlcN indicating changes in DNA replication. Mean intensity normalized to 1 at 24 hpi. **B)** Area under the

curve for three replicas of each condition shown in A) calculated in Prism. Statistical analysis: t-test with Welch´s correction.
(TIF)

**S8 Fig. Western blot analysis shows no increased tubulin, centrin, or aldolase abundance in Slp knock down cells.** A) Synchronized Slp parasites +/- GlcN were harvested at 24, 30 or 36hpi, and SDS-PAGE of $2*10^7$, $1*10^7$ or $2*10^6$ parasites per lane respectively was performed. After blotting, the blot was cut at 25 kDa. The upper part was incubated with rabbit-anti-aldolase and mouse anti-tubulin primary antibodies, and anti-rabbit 680RD and anti-mouse 800CW secondaries, while the lower part was incubated with rabbit anti-Centrin3 primary and anti-rabbit 800CW secondary antibodies. Band at 50 kDa corresponds to the molecular weight of tubulin, a lower unspecific band is marked as (*). Band at 37 kDa and 20 kDa correspond to aldolase and centrin size respectively. Unspecific band was not included in measurement B) Quantification of aldolase protein signal after correcting for equal parasite number. C) Quantification of centrin protein level after correcting to equal parasite number, normalized to aldolase signal. Slp -GlcN (blue). Slp +GlcN (red). All means generated from three biological replicates. Statistical analyses: t-test with Welch´s correction. ns: $p>0.05$.
(TIF)

**S9 Fig. Microtubule levels are already increased in the first mitotic spindle of PfSlp knock down parasites.** Movies of SPY555-Tubulin labelled 3D7 +GlcN and Slp +GlcN treated cells acquired in the same imaging session using identic settings. For quantification the first timeframe showing a defined mitotic spindle was selected and the relative microtubule signal intensity was quantified using average intensity projections of image slices containing the spindle microtubule signal. Means generated from three independent imaging sessions (N = 3). SD is shown. Statistical analysis by t-test with Welch´s correction. ***: $p<0.0001$.
(TIF)

**S1 Table. Mass spectrometry analysis of Centrin Co-IP reveals two specific interaction partners.** All proteins identified from the non-crosslinked PfCentrin1-GFP co-immunoprecipitation samples are listed with peptide counts. Only Centrin-1, Centrin-3, and PfSlp (highlighted in green) were not detectable in a GFP only control pull down done with the same protocol used in a previous study by Balestra et al. 2021 [50].
(PDF)

**S2 Table. List of primers used in this study.**
(PDF)

**S3 Table. Antibodies and dyes used in this study. Starred (*) indicates dilutions for IFAs imaged by confocal microscopy; for STED, those antibodies were used at 1:200.**
(PDF)

**S1 Movie. Normal mitotic spindle formation and extension dynamics in Slp parasite.** Super-resolution time-lapse confocal microscopy of Slp blood stage parasite -GlcN labelled with microtubule live cell dye SPY555-Tubulin (magenta) and acquisition of transmission mode (grey) undergoing the first few mitotic divisions. Images were acquired with Zeiss Airyscan detector and processed for improved resolution. Maximum projections are shown. Image dimensions are 10 x 10 μm.
(AVI)

**S2 Movie. Normal mitotic spindle formation and extension dynamics in wild type parasite.** Super-resolution time-lapse confocal microscopy of 3D7 wild type blood stage parasite +GlcN

labelled with microtubule live cell dye SPY555-Tubulin (magenta) and acquisition of transmission mode (grey) undergoing the first few mitotic divisions. Images were acquired with Zeiss Airyscan detector and processed for improved resolution. Maximum projections are shown. Image dimensions are 10 x 10 µm.
(AVI)

**S3 Movie. Mitotic spindle extension is strongly delayed in PfSlp knock down.** Super-resolution time-lapse confocal microscopy of Slp blood stage parasite +GlcN labelled with microtubule live cell dye SPY555-Tubulin (magenta) and acquisition of transmission mode (grey) attempting the first mitotic spindle extension. Images were acquired with Zeiss Airyscan detector and processed for improved resolution. Maximum projections are shown. Image dimensions are 10 x 10 µm.
(AVI)

**S4 Movie. Normal mitotic spindle formation and nuclear replication dynamics in Slp parasite.** Super-resolution time-lapse confocal microscopy of Slp blood stage parasite -GlcN labelled with microtubule live cell dye SPY555-Tubulin (magenta) and 5-SiR-Hoechst (cyan) acquisition of transmission mode (grey) undergoing the first few mitotic divisions. Images were acquired with Zeiss Airyscan detector and processed for improved resolution. Maximum projections are shown. Please note temporary black bars due to occasionally unavoidable stage drift. Image dimensions are 10 x 10 µm.
(AVI)

**S5 Movie. Impaired nuclear division and normal DNA replication in PfSlp KD parasite.** Super-resolution time-lapse confocal microscopy of Slp blood stage parasite +GlcN labelled with microtubule live cell dye SPY555-Tubulin (magenta) and 5-SiR-Hoechst (cyan) acquisition of transmission mode (grey) attempting the first mitotic division. Images were acquired with Zeiss Airyscan detector and processed for improved resolution. Maximum projections are shown. Please note temporary black bars due to occasionally unavoidable stage drift. Image dimensions are 10 x 10 µm.
(AVI)

**S1 Data. Raw data of figures.**
(XLSX)

## Acknowledgments

We thank: The Infectious Diseases Imaging Platform for imaging support (idip-heidelberg. org). The Instituto Aggeu Magalhães- Fundação Oswaldo Cruz- FIOCRUZ/PE for the collaboration during the sabbatical period of Dr. Tatiany Romão at our Center. PlasmoDB for their Plasmodium Informatics Resources (plasmodb.org). Nicolas Lichti for help with molecular cloning. The MSc students of the Major Infectious Diseases that participated in the practical course on Pathogenic Microorganisms 2021 for generating preliminary imaging data. The excellent service at the proteomic core facility at the Faculty of Medicine of the University of Geneva.

## Author Contributions

**Conceptualization:** Christoph Wenz, Caroline Sophie Simon, Mathieu Brochet, Julien Guizetti.

**Data curation:** Christoph Wenz, Caroline Sophie Simon, Julien Guizetti.

**Formal analysis:** Christoph Wenz, Caroline Sophie Simon, Mathieu Brochet.

**Funding acquisition:** Markus Ganter, Mathieu Brochet, Julien Guizetti.

**Investigation:** Christoph Wenz, Caroline Sophie Simon, Tatiany Patricia Romão, Vanessa Saskia Stürmer, Marta Machado, Natacha Klages, Anja Klemmer, Yannik Voß, Mathieu Brochet.

**Methodology:** Christoph Wenz, Caroline Sophie Simon, Tatiany Patricia Romão, Vanessa Saskia Stürmer, Marta Machado, Natacha Klages, Anja Klemmer, Yannik Voß.

**Project administration:** Julien Guizetti.

**Resources:** Caroline Sophie Simon, Tatiany Patricia Romão, Yannik Voß.

**Supervision:** Markus Ganter, Mathieu Brochet, Julien Guizetti.

**Validation:** Caroline Sophie Simon.

**Visualization:** Christoph Wenz, Caroline Sophie Simon, Marta Machado, Julien Guizetti.

**Writing – original draft:** Christoph Wenz, Julien Guizetti.

**Writing – review & editing:** Christoph Wenz, Caroline Sophie Simon, Markus Ganter, Mathieu Brochet, Julien Guizetti.

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
