## [Decision Letter · Decision Letter 0]

11 Mar 2023

Dear Dr. Guizetti,

Thank you very much for submitting your manuscript "An Sfi1-like centrin-interacting centriolar plaque protein affects nuclear microtubule homeostasis." for consideration at PLOS Pathogens. As with all papers reviewed by the journal, your manuscript was reviewed by members of the editorial board and by several independent reviewers. The reviewers appreciated the attention to an important topic. Based on the reviews, we are likely to accept this manuscript for publication, providing that you modify the manuscript according to the review recommendations.

Please pay attention to comments by Reviewer 2 while revising the manuscript, especially the comment about the discrepancies in analysis of images and Western blots and qPCR. Segregating the results in separate sections would also improve the manuscript.  Reviewers 1 and 3 also suggest several changes in the manuscript that should be appropriately addressed in your revision.

Sincerely,

Akhil B. Vaidya

Guest Editor

PLOS Pathogens

Margaret Phillips

Section Editor

PLOS Pathogens

Kasturi Haldar

Editor-in-Chief

PLOS Pathogens

orcid.org/0000-0001-5065-158X

Michael Malim

Editor-in-Chief

PLOS Pathogens

orcid.org/0000-0002-7699-2064

Reviewer Comments (if any, and for reference):

Reviewer's Responses to Questions

**Part I - Summary**

Reviewer #1: The study led by Wenz and Simon identified the malaria parasite homolog of Sif1, a centrin-binding protein. They demonstrated that Sif1 interacts with centrin 1 and localizes at the cytosolic compartment of the microtubule organizing center in the Plasmodium falciparum blood-stage parasite. To address Sfi1 function during the parasite intraerythrocytic replication, they generated a conditional knockdown parasite line using the glms Ribozyme system. Next, they employed a combination of super-resolution and live microscopies to demonstrate the critical role of Sif1 in intranuclear homeostasis of tubulin, proper DNA segregation, and parasite growth. The study is well conducted with adequate controls and biological replicates, resulting in conclusive new findings on the atypical Plasmodium cell division mode. The three reviewers from Review commons raised most of my concerns from the preprint version, and the authors' responses satisfied me. I agree with Reviewer 3 comments regarding the additional experiment to test whether Sfi1 is a checkpoint factor that would have strengthened the manuscript by bringing more mechanistic to the more phenotypical description of Sfi1 KD. Lastly, I have a few minor comments for the authors to address before the final publication.

Reviewer #2: The resubmission of the manuscript by Wenz et al. is devoted to the role of the Plasmodium ortholog of the yeast half-bridge protein Sfi in the parasite cell division. The role of this centrin-interacting protein had not been examined in Plasmodium sp., and, according to the evidence presented in the manuscript, this factor may regulate the first karyokinetic event of the multinuclear division. Although the authors supplemented the revised manuscript with new findings, the study remains limited in scope and needs more experimental rigor. As such, it requires substantial work to support the model of the PfSlp1 function in Plasmodium.

Reviewer #3: Plasmodium falciparum parasites undergo several rounds of asynchronous nuclear divisions to produce daughter cells. This process is controlled by the centriolar plaque, a non-canonical centrosome that functions to organize intranuclear spindle microtubules. The organization and composition of this microtubule organizing center is not well understood. Here, Wenz et al. identify a novel centrin-interacting protein, PfSlp, that, following knockdown, leads to fewer daughter cells and aberrant intranuclear microtubule homeostasis and organization.

Wenz et al. identify PfSlp via co-immunoprecipitation of P. falciparum 3D7 strain with an episomally expressed PfCen1-GFP, noting PfSlp as a gene of interest based on the presence of several centrin-binding motifs. The authors go forward to generate a transgenic 3D7 strain, equipping PfSlp with GFP and glmS ribozyme, to localize and evaluate the function of PfSlp in asexual blood stage parasites. PfSlp appears to, using immunofluorescence and STED microscopy, localize to the outer centriolar plaque in schizonts, based on its colocalization with PfCen3. Moreover, PfSlp appears to interact with PfCentrin as evident by western blot analysis following a reciprocal IP using anti-GFP on Slp-GFP parasites. The authors show, utilizing the inducible glmS ribozyme knockdown system, that PfSlp is required for proper parasite growth, noting a replication defect following addition of GlcN. This defect is noted to cause a delay in the initiation of nuclear division, or schizogony. Analysis of intranuclear microtubule dynamics reveal abnormal microtubule organization, specifically an increase in nuclear microtubule abundance and length following PfSlp knockdown. Together, these findings characterize the role of a novel protein, PfSlp, that contributes to nuclear tubulin homeostasis and organization during schizogony.

Major comments:

The major claims made by Wenz et al. are convincing with the data provided. The changes made are satisfactory in response to reviewer comments. Conclusions made about PfSlp and centrins are interesting and strengthened by the addition of the reciprocal IP with Slp parasites. The data presented is clear and biological replicates and proper statistics are present. The discussion of cell cycle checkpoint is interesting and conclusions regarding the impact of this work on this question in the field is not overstated

**Part II – Major Issues: Key Experiments Required for Acceptance**

Reviewer #1: None

Reviewer #2: There is a significant discrepancy between the analysis of individual parasites (movies/images) and the bulk (WB, qPCR). For example, there is no change in tubulin expression by WB, while the movies show the multiplication of the tubulin dots in Slp1 expressing parasites. This is likely the result of inefficient PfSlp1 knockdown, which was brought to the anthers’ attention in the previous submission. A 55% mRNA reduction after 73h with GlcN is not an efficient knockdown for looking at the target protein function. Furthermore, the Slp1 tagging/expression should be demonstrated at the protein level. Although the size of the protein is an understandable obstacle, it is a poor excuse for the lack of evidence. There are technologies to enhance the signal (spaghetti monster epitopes, large protein resolution PAGE). In line with this concern, how do you set up experiments if the maximum effect is at 73h and the lytic cycle of the parasite is 48h?

The data interpretation/analysis is questionable. The main conclusion of the PfSlp1 role in mitosis is founded on the results shown in Fig. 4. However, if the graph in Fig. 4C was made of the movies 4-5, then quantifications of the DNA replication do not match what is in video files. At the late time points (~600 min), non-treated parasites have significantly brighter Hoechst intensity (and multiple nuclei) than GlcN-treated parasites (one nucleus). At the same time, the lines of both experimental sets are nearly merged on the graph. Also, the number of tubulin dots is not the proper representation of the number of nuclei: the same nucleus in pre-mitosis has 1 dot, in metaphase 2 dots, and back to 1 dot in anaphase and telophase.

The manuscript needs better organization. At times the writing is redundant and confusing, and the results need to be segregated into sections.

Reviewer #3: The following are areas that need to be addressed:

• Line 82: You say “cytoplasmic microtubules are absent in schizonts.” This isn’t true. There is a single spine of subpellicular microtubules in later stage schizonts and fully formed merozoites (see reference). These are widely observed in the field.

o Harding, C. R. & Frischknecht, F. The Riveting Cellular Structures of Apicomplexan Parasites. Trends Parasitol. 36, 979–991 (2020).

• Line 226-227 AND Figure 3C: You observe tubulin protrusions but do not quantify the frequency at which you observe this in Slp + GlcN parasites compared to your controls. Sometimes parasites just look weird, and quantification of this phenotype will strengthen your claim.

• Line 369-371: You say “the centriolar plaque...raises the possibility that this specialized nuclear pore.” This sentence suggests that the centriolar plaque is a nuclear pore – is this what you mean? If so, please give more explanation and clarify.

**Part III – Minor Issues: Editorial and Data Presentation Modifications**

Reviewer #1: Line 81: The authors wrote that " while cytoplasmic microtubules ae absent in schizonts" knowing that subpellicular microtubules assemble from the apical ring in the parasite cytoplasm during schizogony, the authors must clarify that they meant that no cytoplasmic microtubules are nucleated by the CP.

Line 86: As Simon et al 2021 demonstrated and used, NHS-ester is a CP marker in parasite post expansion microscopy. Therefore I suggest the authors to write" the only currently known extranuclear CP marker in non-expanded parasites".

I found the data in sup fig 8 essential to support the functional role of Sfi1 in Tubulin homeostasis and would move the data to the main figure 3.

In the discussion section, the authors claimed in line 369: " The positioning of PfSfi1 close to the neck of the centriolar plaque" What are the data supporting this claim and could they define what is the neck of the centriolar plaque. In Simon et al 2021 they refer the CP as a hourglass shape structure and it is unclear what is the CP neck.

Reviewer #2: It is unclear why some of the data is included. Lines 185-195 and Fig 2D show no difference between the parent and the PfSlp1 strain treated with GlsN. It does not add to the story but instead makes the story unfocused.

Reviewer #3: Minor comments:

• Line 42: Replace “begin” with “beginning”

• Line 45: Replace “stage” with “stages”

• Line 142: You say “Upon transition into schizogony the schizont stage late trophozoites develop a hemispindle in their nucleus of which about half carry a centrin signal.” Include a comma, as follows “the schizont stage late, trophozoites develop”

• Figure 1A: Your schematic depicting the first nuclear division labels a 1N parasite (with a hemispindle assembled) as a “schizont.” Schizonts, to our understanding, are parasites with 3+ nuclei (see reference).

o Delahunt, C., Horning, M. P., Wilson, B. K., Proctor, J. L. & Hegg, M. C. Limitations of haemozoin-based diagnosis of Plasmodium falciparum using dark-field microscopy. Malar. J. 13, 147 (2014).

• Figure 1D: Avoid the word “zoom” when referring to your STED images. It is confusing and may lead readers to think these are digital zoom-ins of your confocal images rather than separate STED images.

• Line 277: You mention a “mitotic spindle phase duration” but do not write the value in the text as it is written. Would be nice to include.

• Figure 4D: Include color labels in your figure legend.

PLOS authors have the option to publish the peer review history of their article (what does this mean?). If published, this will include your full peer review and any attached files.

Reviewer #1: **Yes: **Sabrina Absalon

Reviewer #2: No

Reviewer #3: No

Figure Files:

Data Requirements:

Reproducibility:

References:

---

## [Editor Report · Decision Letter 1]

28 Mar 2023

Dear Dr. Guizetti,

We are pleased to inform you that your manuscript 'An Sfi1-like centrin-interacting centriolar plaque protein affects nuclear microtubule homeostasis.' has been provisionally accepted for publication in PLOS Pathogens.

Best regards,

Akhil B. Vaidya

Guest Editor

PLOS Pathogens

Margaret Phillips

Section Editor

PLOS Pathogens

Kasturi Haldar

Editor-in-Chief

PLOS Pathogens

orcid.org/0000-0001-5065-158X

Michael Malim

Editor-in-Chief

PLOS Pathogens

orcid.org/0000-0002-7699-2064
---

## [Editor Report · Acceptance letter]

27 Apr 2023

Dear Dr. Guizetti,

We are delighted to inform you that your manuscript, "An Sfi1-like centrin-interacting centriolar plaque protein affects nuclear microtubule homeostasis.," has been formally accepted for publication in PLOS Pathogens.

Best regards,

Kasturi Haldar

Editor-in-Chief

PLOS Pathogens

orcid.org/0000-0001-5065-158X

Michael Malim

Editor-in-Chief

PLOS Pathogens

orcid.org/0000-0002-7699-2064